

# Constraining the geotherm beneath the British Isles from Bayesian inversion of Curie depth: integrated modelling of magnetic, geothermal and seismic data

Ben Mather[1] and Javier Fullea[1]

[1]Dublin Institute for Advanced Studies, School of Cosmic Physics, 5 Merrion Square, Dublin 2, Ireland

**Correspondence:** Ben Mather (bmather@cp.dias.ie)

**Abstract.** Curie depth offers a valuable constraint on the thermal structure of the lithosphere, based on its interpretation as
the depth to $580\,^\circ\text{C}$, but current methods underestimate the range of uncertainty. We formulate the estimation of Curie depth
within a Bayesian framework to quantify its uncertainty across the British Isles. Uncertainty increases exponentially with Curie
depth but this can be moderated by increasing the size of the spatial window taken from the magnetic anomaly. The choice
of window size needed to resolve the magnetic thickness is often ambiguous, but based on our chosen spectral method, we
determine that significant gains in precision can be obtained with windows sizes 15–30 times larger than the deepest magnetic
source. Our Curie depth map of the British Isles includes a combination of window sizes: smaller windows are used where the
magnetic base is shallow to resolve small-scale features, and larger window sizes are used where the magnetic base is deep in
order to improve precision. On average, the Curie depth increases from Laurentian crust ($22.2 \pm 5.3$ km) to Avalonian crust
($31.2\pm9.2$ km). The temperature distribution in the crust, and associated uncertainty, was simulated from the ensemble of Curie
depth realisations assigned to a lower thermal boundary condition of a crustal model (sedimentary thickness, Moho depth, heat
production, thermal conductivity), constructed from various geophysical and geochemical data sets. The uncertainty of the
simulated heat flow field substantially increases from $\pm\,10$ mW m$^{-2}$ for shallow Curie depths $\sim 15$ km to $\pm\,80$ mW m$^{-2}$
for Curie depths $> 40$ km. Surface heat flow observations are concordant with the simulated heat flow field except in regions
that contain igneous bodies. Heat flow data within large batholiths in the British Isles exceed the simulated heat flow by
$\sim 25\,\text{mW}\,\text{m}^{-2}$ as a result of their high rates of heat production (4–6 $\mu$W m$^{-3}$). Conversely, heat refraction around thermally
resistive mafic volcanics and thick sedimentary layers induce a negative heat flow misfit of a similar magnitude. A northward
thinning of the lithosphere is supported by shallower Curie depths on the northern side of the Iapetus Suture, which separates
Laurentian and Avalonian terranes. Cenozoic volcanism in Northern Britain and Ireland has previously been attributed to a
lateral branch of the proto-Icelandic mantle plume. Our results show that high surface heat flow ($> 90$ mW m$^{-2}$) and shallow
Curie depth ($\sim 15$ km) occur within the same region, which supports the hypothesis that lithospheric thinning occurred due to
the influence of a mantle plume. That the uncertainty is only $\pm$ 3–8 km in this region, demonstrates that Curie depths are more
reliable in hotter regions of the crust where the magnetic base is shallow.



## 1 Introduction

Magnetic data are, along with gravity anomalies, the most commonly available geophysical observables for subsurface imaging of the Earth. Surveys of the magnetic anomaly capture, among other features, the contribution of various magnetic minerals to the surface field recorded by magnetometers. Ferromagnetic minerals retain their magnetism until they reach their Curie temperature. Magnetite is the most prevalent magnetic mineral, in terms of susceptibility and quantity, and has a Curie temperature of $580\,°C$. Therefore, the Curie depth is often interpreted as the depth to the $580\,°C$ isotherm. For this reason, Curie depth offers a valuable constraint to estimate the geothermal heat flow.

Various methods have been proposed to estimate Curie depth from a spectral analysis of the magnetic anomaly, which have been successfully applied to many regions of the Earth, but often under-represent the degree of uncertainty within each Curie depth estimate. We propose a Bayesian approach where Curie depth is expressed in probabilistic terms. We apply this methodology to the British Isles, to quantify the degree of uncertainty in Curie depth, and use this as a boundary condition to compute the temperature distribution in the crust, integrating a stratified geological model constrained by heat production, seismic and surface heat flow data. We validate our prediction of geothermal heat flow across the British Isles against surface heat flow data to examine the variations and controls on crustal heat flow.

### 1.1 Geological and tectonic context

The British Isles comprise Laurentian and Avalonian lithosphere that were brought together at the closure of the Iapetus Ocean at 420 Ma (Chew and Strachan, 2014) (Figure 1). The Iapetus Suture Zone (ISZ) is an area of deformation along which continental accretion took place during the Caledonian Orogeny (Soper et al., 1992). Seismic studies identify the ISZ as a thickened mid-crust along a 60 km-wide zone beneath central Ireland and northern England (Klemperer et al., 1991; Brewer et al., 1983). The major exposed Caledonian granite batholiths were emplaced during widespread magmatism from the end of the Silurian to the Devonian (Brown et al., 2008). Extension associated with sea-floor spreading in the North Atlantic during the Cretaceous formed sedimentary basins off the west coast of Ireland (e.g. O'Reilly and Griffin, 2010). Widespread extrusion of flood basalts during the Cenozoic, possibly associated with a branch of the Icelandic mantle plume (Landes et al., 2007), formed the North Atlantic Igneous Province between Northern Ireland and Scotland.

The $V_p$ structure of the British Isles generally support a uniformly thick crust in Avalonia of 32–35 km (Landes et al., 2005; Davis et al., 2012), which thins north of the ISZ from 35 to 25 km in northwest Scotland and Ireland (Tomlinson et al., 2006; Licciardi et al., 2014). The Laurentian side of the British Isles has experienced significant igneous activity, initially during Caledonian emplacement of granite batholiths and later due to extrusion of mafic volcanics. High radiogenic heat production rates (4 to 8 $\mu$W m$^{-3}$) have been inferred in the Laurentian terrane of Ireland and Scotland (Webb et al., 1987; Mather et al., 2018). In contrast, the comparatively fewer granitic bodies outcroping in the Avalonian crust are associated with lower heat production rates (3.5–5.5 $\mu$W m$^{-3}$ Webb et al., 1987).



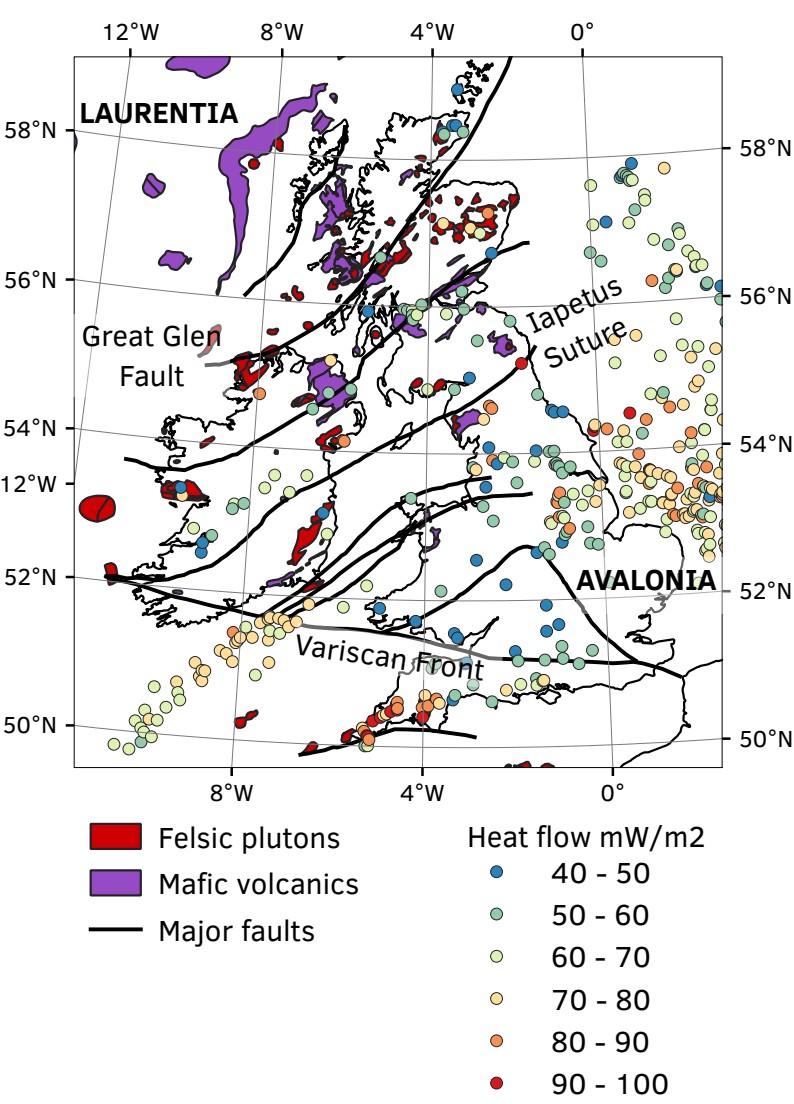

**Figure 1.** Igneous outcrop and major geological structures across the British Isles. Surface heat flow data are compiled from Mather et al. (2018) and Pollack et al. (1993).





## 2 Methodology

We apply probabilistic techniques to quantify the uncertainty of Curie depth (interpreted as the $580\,^{\circ}\mathrm{C}$ isotherm) estimated from magnetic anomaly and the associated surface heat flow. In particular, we cast spectral methods to compute Curie depth within a Bayesian framework and outline a efficient procedure to map Curie depth minimising its uncertainty. The Curie depth is then used as the lower boundary condition of a 3D Cartesian mesh, over which we solve the steady-state heat equation, for a given set of thermal parameters, in order to estimate the surface heat flow and its uncertainty.

### 2.1 Spectral methods to compute Curie depth

Most methods to estimate Curie depth relate the spectrum of magnetic anomalies and the depth of magnetic sources by transforming the data into the Fourier domain, which is computed over a square window of the magnetic anomaly. Depth to the bottom of magnetic sources is estimated from the slope of the radial power spectrum, $\Phi(k)$,

$$\Phi(\mathbf{k}) \propto k^{-\beta} \tag{1}$$

where $\mathbf{k}$ is the wave number in Cartesian space, $k = |\mathbf{k}|$ is its norm, and $\beta$ is the fractal parameter of magnetisation. Originally, Tanaka et al. (1999) analysed the slope of the radial power spectrum at high and low wave numbers to estimate the top and bottom of magnetic sources, respectively. Their method has been applied to many regions worldwide with small modifications (Li et al., 2009; Li, 2011; Salem et al., 2014; Salazar et al., 2017; Martos et al., 2017). However, the difficulty lies in choosing which segments of $\Phi$ over which to calculate the depth of magnetic sources. Often, this is inspected visually and is highly subjective to the interpreter.

Alternatively, Bouligand et al. (2009) offered an analytic expression to the radial power spectrum that expanded on the formulation of Maus et al. (1997),

$$\Phi(k) = C - 2kz_t - k\Delta z - \beta \ln(k)$$
$$+ \left[ -k\Delta z + \ln \left( \frac{\sqrt{\pi}}{\Gamma(1 + \frac{\beta}{2})} \left( \frac{\cosh(k\Delta z)}{2} \Gamma \left( \frac{1+\beta}{2} \right) - K_{\frac{1+\beta}{2}}(k\Delta z) \left( \frac{k\Delta z}{2} \right)^{\frac{1+\beta}{2}} \right) \right) \right] \tag{2}$$

where the shape of the power spectrum is controlled by four variables: $\beta$ – a fractal parameter, $z_t$ – the top of magnetic sources, $\Delta z$ – the thickness of the magnetic layer, and $C$ – a field constant. The Curie depth is found by summation of $z_t$ and $\Delta z$. Some combination of these parameters should produce a curve that fits the radial power spectrum computed from a fast-Fourier Transform (FFT) of the magnetic anomaly. Computing Curie depth in this manner offers a significant advantage over other methods because the fit between analytic and computed power spectrum can be quantified.

### 2.2 The inverse problem of Curie depth

Here we consider the analytic expression from Eq 2 to compute the power spectra from the magnetic anomaly and cast it within a Bayesian framework, where information on input parameters are represented in probabilistic terms. The inverse solution is





given by the *a posteriori* probability function, $P(\mathbf{m}|\mathbf{d})$, which is proportional to the product of the likelihood function $P(\mathbf{d}|\mathbf{m})$
and the *a priori* probability $P(\mathbf{m})$,

$$P(\mathbf{m}|\mathbf{d}) \propto P(\mathbf{d}|\mathbf{m}) \cdot P(\mathbf{m}) \tag{3}$$

The likelihood function is the probability of reproducing the data $\mathbf{d}$ given a particular model $\mathbf{m}$, and the *a priori* probability
is prior knowledge about the model before assimilating the data. In this case, the model parameters correspond to the four
unknowns in Equation 2 and the data is the radial power spectrum computed from the magnetic anomaly, $\Phi_d$, thus $P(\mathbf{m}|\mathbf{d}) =$
$P(\beta, z_t, \Delta z, C|\Phi_d)$. The posterior probability can be evaluated through an objective function, $S(\mathbf{m})$, which compares the misfit
between data and prior information, $\mathbf{m}_p$,

$$P(\mathbf{m}|\mathbf{d}) = A \exp(-S(\mathbf{m})) \tag{4}$$

where $A$ is a constant. The maximum *a posteriori* (MAP) estimate is obtained by minimising the $\ell_p$-norm objective function if
data and prior information are both uncorrelated,

$$S(\mathbf{m}) = \frac{1}{s} \sum_i \frac{|g^i(\mathbf{m}) - \mathrm{d}^i|^s}{(\sigma_\mathrm{d}^i)^s} + \frac{1}{r} \sum_j \frac{|\mathrm{m}^j - \mathrm{m}_p^j|^r}{(\sigma_p^j)^r} \tag{5}$$

where $\mathbf{g}$ is the forward operator, which is the prediction of observations from the model, $\mathbf{m}$. This is simply the calculation of $\Phi$
from $\beta$, $z_t$, $\Delta z$, and $C$ in Equation 2. Gaussian probabilities are assumed for $\Phi$ ($s = 2$ i.e. the $\ell_2$-norm) and uniform priors over
a large range (thus the second term in Equation 5 disappears). The Metropolis-Hastings algorithm is implemented to sample
the posterior using a "random walk" (the algorithm is detailed in Appendix A).
## 2.3 Resolution of model parameters: synthetic tests
The Bayesian framework we have described produces an ensemble of models that sample the posterior density function. Here
we design a synthetic sensitivity test to explore the inversion parameters. As the inversion operates in the spectral domain
a crucial parameter is the size of the spatial window used to transform the spatially distributed magnetic data to the Fourier
domain. A synthetic magnetic anomaly was generated from random fractal noise in three dimensions with known model
parameters ($\beta = 3$, $z_t = 0.305$, and $\Delta z = 10$) from which spatial windows were extracted of different sizes to estimate the
posterior uncertainty (Figure 2a). Magnetic sources that reside deeper in the crust affect longer wavelengths of the magnetic
field. Therefore, in order to retrieve Curie depth, the window size must be large enough to resolve low wave numbers in the
radial power spectrum (Figure 2b). Intuitively, fewer points in the low frequency part of the radial power spectrum result in
higher uncertainty, however, the relationship between window size and Curie depth uncertainty is non-linear (Figure 2c). There
are significant gains in precision for window sizes up to 200 km, and smaller improvement thereon. This non-linear relationship
can be exploited to minimise the uncertainty and the computational time. Bouligand et al. (2009) estimate that window size
should be 10–15 times larger than the depth extent of magnetic sources. Our results suggest that for magnetic sources in the
depth range 10-40 km and uncertainties of 5 km the window size should be closer to 15–30 times the depth extent of magnetic
sources.



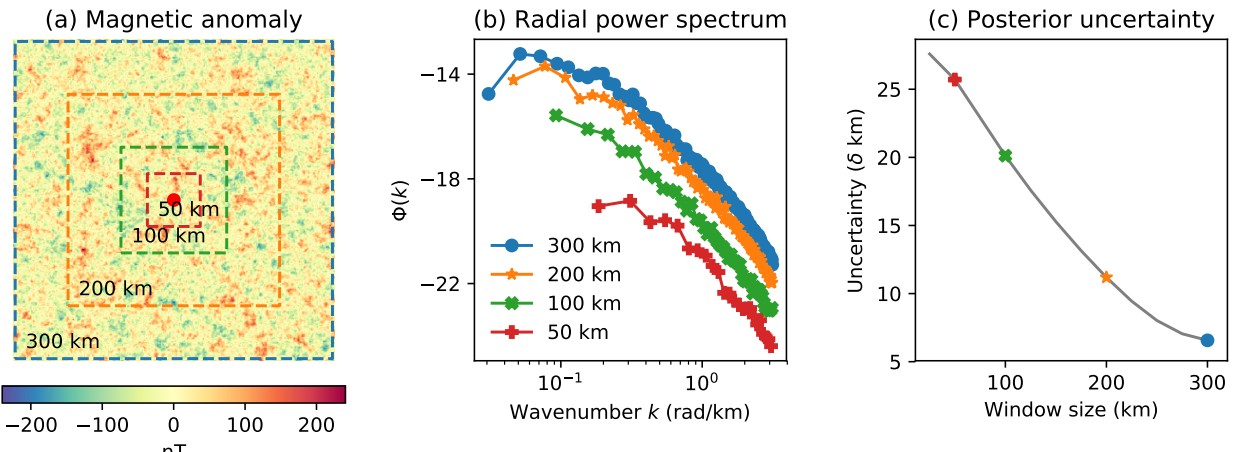

**Figure 2.** The relationship between window size and the uncertainty of Curie depth estimates ($z_t + \Delta z$) for a synthetic model. (a) Synthetic magnetic anomaly generated using the parameters: $\beta = 3$, $z_t = 0.305$, and $\Delta z = 10$; (b) radial power spectrum for different window sizes; (c) uncertainty of Curie depth estimates as a function of window size.

In addition to window size, retrieval of accurate Curie depth values is also complicated by a strong correlation between $\beta$
and $\Delta z$, which both control the slope of the radial power spectrum at low wave numbers (Figure 3b). The variation in these
parameters may be reduced with *a priori* information related to $\beta$ or $z_t$, however, in most circumstances these parameters are
completely unknown.
**2.4    Modelling geothermal heat flow**
Geothermal heat flow can be determined by modelling a geotherm from the surface of the crust to the Curie depth. A geotherm
is modelled by solving the steady-state heat equation,

$$\nabla(\lambda(\nabla \mathbf{T})) = -\mathbf{H} \qquad (6)$$

where $\lambda$ is the thermal conductivity and $H$ is the rate of heat production. Equation 6 is solved in 3D numerically with top and
bottom Dirichlet boundary conditions set to the mean annual surface temperature and Curie temperature of magnetite (12 and
$580\,°\text{C}$, respectively). The top of the model is the surface topography (Amante, 2009), and the base of the model is adjusted
to the Curie depth from the inversion of magnetic anomaly as outlined in sections 2.1 and 2.2. To account for the decrease in
thermal conductivity with temperature, $\lambda$ is iteratively updated according to the relationship of Durham et al. (1987),

$$\lambda(\mathbf{T}) = 2.26 - \frac{618.241}{\mathbf{T}} + \lambda_0 \left( \frac{355.576}{\mathbf{T}} - 0.30247 \right) \qquad (7)$$

where $\lambda_0$ is the thermal conductivity at the surface. Surface heat flow, $q_s$, is calculated from the product of thermal conductivity
and temperature gradient in the topmost part of the model once the temperature solution has converged after several iterations
($|\mathbf{T}_n - \mathbf{T}_{n-1}| < 10^{-12}$).





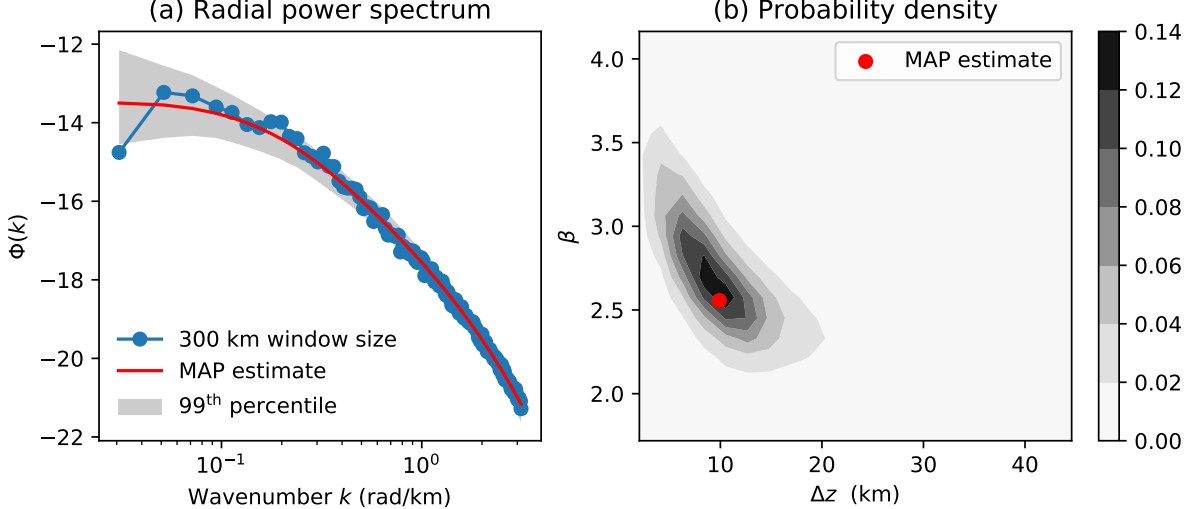

**Figure 3.** Ensemble of radial power spectra from a MCMC simulation. (a) Radial power spectrum computed over a 300 km window of a synthetic magnetic anomaly – shaded region indicates the 99[th] percentile of all models in the ensemble and the red curve is the MAP estimate; (b) joint posterior probability density for $\beta$ and $\Delta z$ – the red dot indicates MAP estimate.

## 3   Results

Maps of Curie depth in the study region were taken from the magnetic anomaly of the British Isles that was extracted from the EMAG2 global compilation (Figure 4). While its resolution of 2-arc-minutes (approximately 4 km) is low compared to some regional surveys, the global extent permits Curie depth estimates over very large window sizes. Geographical coordinates were transformed to a local Cartesian projection prior to applying the Fourier transform for the radial power spectrum to be in units of radians per kilometre. We repeated the Markov Chain Monte Carlo (MCMC) simulation for 200 to 800 km windows of the magnetic anomaly at 25 km increments. The posterior distribution is best fit by a skewed normal probability distribution function where the MAP estimate is the mode of the distribution and the uncertainty is its standard deviation.

The implemented map generation algorithm starts at the MAP estimate for the largest window size (800 km) at every point in the model domain and iteratively reduces at 25 km increments until the standard deviation of the posterior exceeds that of the maximum window size. All other Curie depth estimates using smaller or larger window sizes for a given point are rejected. As a consequence, the inverted Curie depth map contains a combination of window sizes: smaller windows are used where the magnetic base is shallow to resolve small-scale features, and larger windows are used where the magnetic base is deep in order to improve precision (Figure 5). While Curie depth was estimated at 200 km, no regions were extracted from this window size due to the high uncertainty.





## 3.1 Curie depth of the British Isles and its uncertainty

A broad linear increase in the MAP estimate of Curie depth is observed from north to south of the British Isles (Figure 6a). On average, Curie depth increases from $22.2 \pm 5.3$ km in Laurentian crust to $31.2 \pm 9.2$ km in Avalonian crust, concordant with a thinning of the crust north of the ISZ (Tomlinson et al., 2006; Licciardi et al., 2014). In Ireland, the Curie depth is deeper in the southwest which points to a cooler geotherm in line with the comparatively higher electrical resistivity found in a magnetotelluric survey southwest of the ISZ (Rao et al., 2014). In contrast with low electrical resistivity (for which there are various possible causes), high electrical resistivity can only be linked to dry and cold rocks (e.g. Pommier and Garnero, 2014; Khan, 2016; Fullea, 2017). A similar SW–NE pattern for shear wave velocity anomaly, with positive anomalies in the southwest, is present in a European seismic tomography model based on surface- and S-waveforms (Legendre et al., 2012). A European scale adjoint waveform tomography model shows fast upper mantle velocity in the eastern English and southwestern Irish margins, whereas slow upper mantle is imaged in the S-W English peninsula (Zhu et al., 2015). Mantle seismic velocities are mainly controlled by thermal variations, with high velocities associated with low temperature and vice versa (Goes et al., 2000; Trampert, 2004; Cammarano and Romanowicz, 2007; Afonso et al., 2013; Fullea et al., 2012). Integrated geophysical-petrological modelling of gravity, elevation and seismic data also indicates a thermal lithospheric thinning (20–30 km), which would suggest shallow Curie depth, in the North of Ireland and western Scotland (Fullea et al., 2014; Baykiev et al., 2018).

Curie depth is $< 20$ km in Northern Ireland and Scotland, which is strongly associated with the high spatial density of plutons and mafic volcanics (Figure 1). Curie depth at the northern edge of the Midland Platform in England is also shallow ($\sim 20$ km) despite low surface heat flow in this region. A deepening is observed south of the Variscan Front in Ireland (35–40 km), but does not continue across to southern England.

For the most part, the Curie depth we determine is concordant with previous studies. The global reference model of Li et al. (2017) is also based on the same EMAG2 magnetic anomaly and follows the similar pattern of variation we observe in the British Isles, albeit at lower resolution. Their method is adapted from Tanaka et al. (1999) and does not consider the estimated uncertainty of Curie depth. In Baykiev et al. (2018), the Curie depth (without uncertainty) is determined from forward modelling gravity, seismic and surface elevation data along with the vertically integrated magnetic susceptibility of a simplified global model. Variations in the Curie depth are broadly similar to our results, except the amplitude of variation is substantially reduced to within 25–35 km depth.

We find that quantifying Curie depth within a Bayesian framework adds significant insight into the thermal structure of the crust. The ensemble of model simulations reveal that uncertainty correlates with Curie depth (Figure 6b). For Curie depths $\sim 15$ km this equates to $\pm 4$ km uncertainty, but deeper Curie depths have a significantly higher uncertainty. This is controlled by window size – larger windows of the magnetic anomaly are required to resolve deeper magnetic thickness (Figure 7a). On average, the Curie depth of the British Isles is between 10–45 km and requires 200–600 km windows of the magnetic anomaly, however, this extends to 70 km depth in the southwest of Ireland and east of England and requires up to 800 km window sizes. Uncertainty increases exponentially with Curie depth (Figure 7b), which is due to fewer points in the low wavenumber portion of the radial power spectrum. The resulting effect on the posterior is an increasingly positive skew with Curie depth





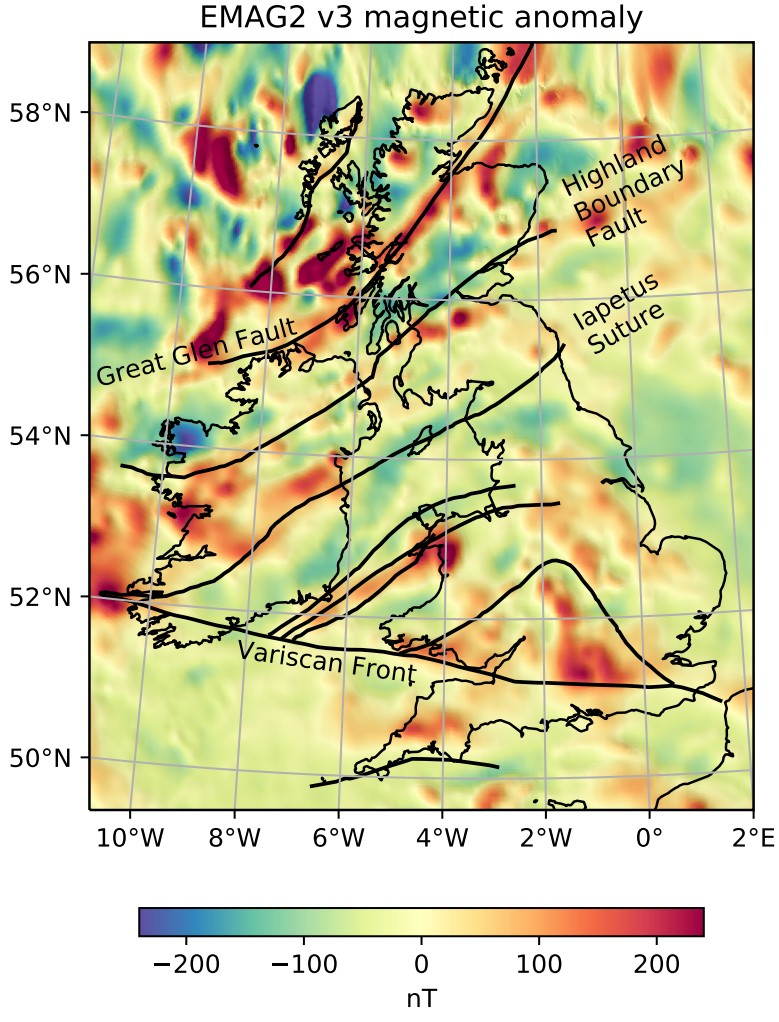

**Figure 4.** Magnetic anomaly map of the British Isles, extracted from the EMAG2 version 3 global compilation (Meyer et al., 2017), and overlain with structural information adapted from Tomlinson et al. (2006).

(Figure 7c), i.e. a longer tail in the upper quartile range of the probability distribution. Very little skew is observed between
10–20 km Curie depth for window sizes $\geq$ 500 km, which indicates the posterior is a Gaussian normal distribution. The degree
of skew increases at a much higher rate for smaller window sizes at Curie depths $>$ 40 km. Only the 800 km window size
exhibits minor skew for magnetic sources deeper than 40 km. These results illustrate how the skew of the posterior distribution
can diagnose window sizes that are insufficient to reliably estimate Curie depth.





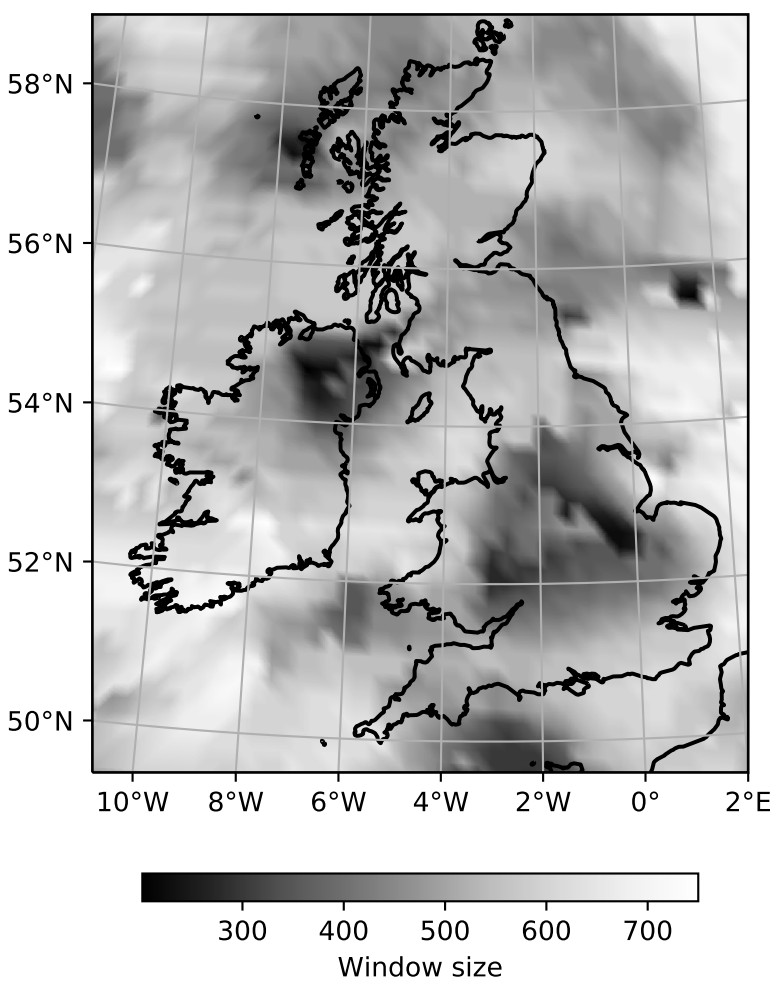

**Figure 5.** Combination of window sizes used to generate the map of Curie depth (Figure 6).





**Figure 6.** MAP estimate of Curie depth in the British Isles and its uncertainty, (a) Curie depth map is computed from 3900 centroids for a lateral resolution of 20 km; (b) Uncertainty is given as one standard deviation from the mean.

**3.2 Heat flow and its uncertainty**
The heat flow distribution over the British Isles is determined by modelling a geotherm from the surface of the crust to the
Curie depth. The top of the model is the surface topography from ETOPO2 (Amante, 2009), set to the mean annual surface
temperature, and the base of the model is adjusted to the Curie depth that was determined for the British Isles. The volume
between these boundaries is divided based on the layers compiled in Baykiev et al. (2018), which contains sedimentary and
Moho (crust-mantle boundary) depth constrained by various geophysical sources (active seismic, receiver functions, gravity
data). The crust is divided into an upper and lower layer, motivated from the well-established theory on the partitioning of
heat-producing elements (e.g. Michaut et al., 2009), where an enriched upper layer $H_s$ (of thickness $h_s$) overlies a relatively





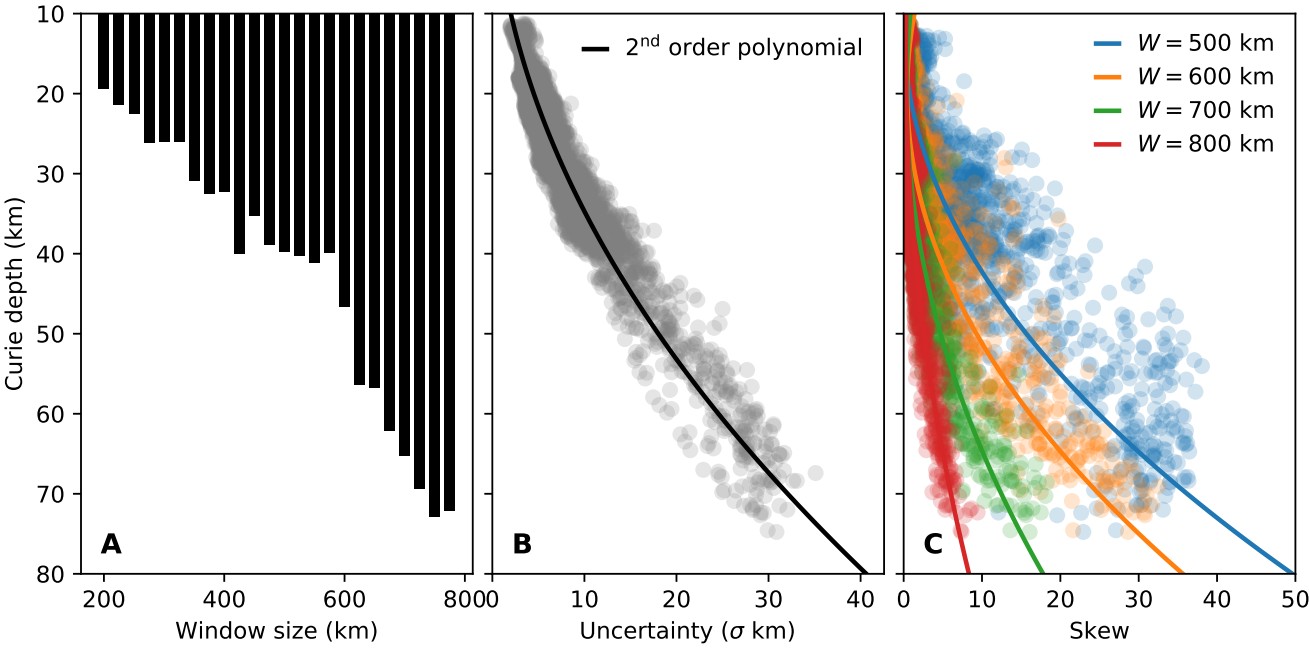

**Figure 7.** Analysis of Curie depth estimates (a) the maximum Curie depth that can resolved for a given window size, (b) the relationship of uncertainty, measured by one standard deviation, with respect to Curie depth, (c) the degree of skew exhibited by the posterior probability distribution as a function of Curie depth for different window sizes – skew is defined here as the mean Curie depth less the median.

depleted lower layer $H_r$ (of thickness $h_r$). The total heat production of the crust is the sum of these two layers,

$$\int_{\text{crust}} H_c \, dh = H_c h_c = H_s h_s + H_r h_r \tag{8}$$

where $h_c$ is crustal thickness. The differentiation index describes the degree of partitioning in the crust,

$$D_I = \frac{H_s}{H_c} \tag{9}$$

where higher $D_I$ indicates that a heat flow province has undergone more significant crustal reworking of heat-producing
elements to the uppermost crust. The vertical differentiation of heat-producing elements in the crust directly controls the cur-
vature of the geotherm and, thus, depth to the $580\,^{\circ}\text{C}$ isotherm. Thus, depth to the $580\,^{\circ}\text{C}$ isotherm will increase if $H_s \gg H_r$
and $h_s \ll h_r$. These values have been previously determined for British Isles from Mather et al. (2018), where the rates of heat
production are defined separately for the crust in the Laurentian and Avalonian terranes (Table 1). We split the total thickness
of the crust ($h_c$) from Baykiev et al. (2018) into an upper (enriched) and lower (depleted) layer of variable thickness by rear-
ranging Equation 8 to respect the differentiation index across Laurentian and Avalonian crust ($D_I = 2.8$ and $1.5$, respectively).
Therefore, the thickness of the upper (enriched) layer is,

$$h_s = h_c \frac{H_c - H_r}{H_s - H_r} \tag{10}$$



| Symbol | Layer | $\lambda_0$ | $H$ |
|---|---|---|---|
| $h_{\text{sed}}$ | Sediments | 2.2 | 1.5 |
| $h_s$ | Upper enriched layer (Laurentia) | 3.0 | 3.9 |
| $h_s$ | Upper enriched layer (Avalonia) | 3.0 | 1.8 |
| $h_c$ | Lower depleted layer (Laurentia) | 3.0 | 0.8 |
| $h_c$ | Lower depleted layer (Avalonia) | 3.0 | 0.1 |

**Table 1.** Thermal properties assigned to each layer in the British Isles, summarised from Mather et al. (2018). Units of $\lambda_0$ are $\text{W m}^{-1}\text{K}^{-1}$ and $H$ are $\mu\text{W m}^{-3}$.

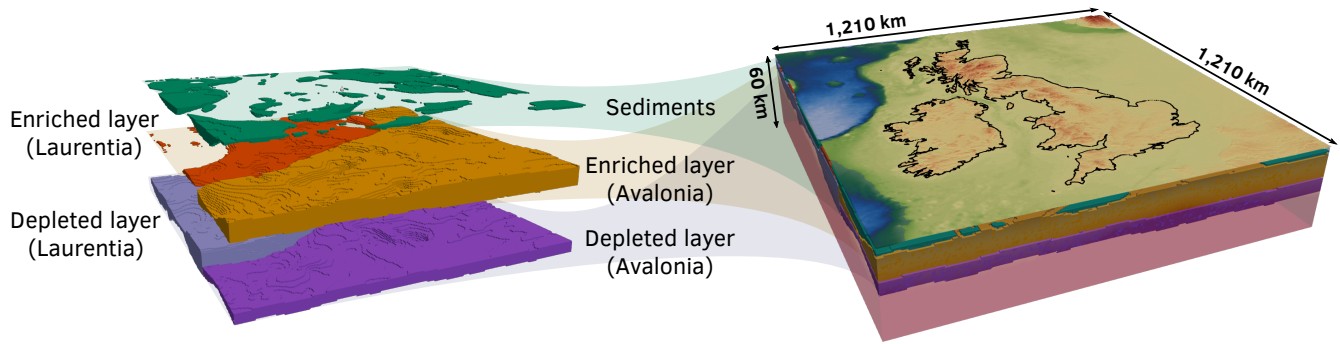

**Figure 8.** Schematic of each layer in the 3D thermal model of the British Isles, subdivided from surfaces compiled in Baykiev et al. (2018). The model is divided by the Iapetus Suture that separates Laurentian and Avalonian crust. The thickness of the upper (enriched) and lower (depleted) crustal layers is determined by rearranging Equation 8 for $h_s$ and $h_r$, respectively, in Laurentian crust ($H_c = 1.4\ \mu\text{W m}^{-3}$) and Avalonian crust ($H_c = 1.2\ \mu\text{W m}^{-3}$).

where $H_s$, $H_r$, and $H_c$ are the rates of heat production for the upper, lower layer and entire crust, respectively. The thermal
properties assigned to each layer in our model are summarised in Table 1.
Our geothermal heat flow map offers very high resolution (approximately 4 km) compared to the spatial density of borehole
measurements, and provides insights into the variation of thermal regimes across the British Isles (Figure 9a). The optimal
surface heat flow, $q_s$, was simulated by fitting the geometry of the lower flux boundary condition to the MAP estimate of
Curie depth. Its uncertainty is determined from randomly perturbing the geometry of the lower boundary condition within the
Curie depth uncertainty to build an ensemble of $q_s$ models and taking the standard deviation (Figure 9b). Here, we present
a probabilistic estimate of surface heat flow based primarily on magnetic data and supported by geothermal observations
and crustal architecture. Generally, heat flow is higher in Scotland and the northernmost parts of Ireland that are of Laurentian
origin. The heat flow contrast on either side of the Iapetus Suture is linked to the crustal partitioning of heat-producing elements,
but there is significant variation of $q_s$ despite the relatively uniform Curie depth in the northern half of the British Isles. Much
of this variation is due to the inclusion of sediments in the model which are thermally insulating ($\lambda_0 = 2.2\,\text{W m}^{-1}\,\text{K}^{-1}$).



**Figure 9.** Surface heat flow in the British Isles and its uncertainty, (a) Surface heat flow overlain with palaeoclimate-corrected heat flow observations compiled from Mather et al. (2018) in Ireland and Pollack et al. (1993) elsewhere – larger circles indicate a higher misfit compared to the simulated heat flow, (b) uncertainty of heat flow determined from an ensemble of model simulations with variable Curie depth, overlain with the locations of heat flow data in white. Circled regions include GG = Galway Granite, PB = Palaeogene Basalt, SWEB = South-West England Batholith, NEB = Northern England Batholith, EHB = Eastern Highlands Batholith.



## 4    Discussion

The simulated heat flow is mostly concordant with heat flow data except for locations above felsic bodies or mafic volcanics. These are not included within the 3D model due to the unconstrained nature of their geometry. Granites can have very high rates of heat production which, when integrated across their thickness, contribute significantly to surface heat flow. This is most prevalent in the south-west England and northern England Batholiths that contain rates of heat production between 3–5 $\mu\mathrm{W\,m^{-3}}$ (Webb et al., 1987) where $+25\,\mathrm{mW\,m^{-2}}$ misfit is observed in comparison to the simulated heat flow. Positive misfit also occurs within the Galway Granite in Ireland and the Eastern Highlands Batholith of Scotland. In contrast, heat refracts around mafic volcanic rocks due to their low thermal conductivity. Heat refraction is most apparent in the northernmost part of Ireland where a ring of high heat flow points are scattered around the rim of Palaeogene basalts with low heat flow in its interior (58 mW m$^{-2}$, Mather et al., 2018). While this effect is clearly visible in the British Isles, in locations where bedrock has not been mapped, either due to poor exposure or lack of geophysical information, using Curie depth to infer surface heat flow can be highly uncertain because short-wavelength features – that significantly influence the heat flow field – cannot be resolved. Where possible, surface heat flow derived from Curie depth should be constrained by crustal geometry and validated against heat flow data from boreholes.

The uncertainty of surface heat flow was estimated from spatially varying the depth of the lower boundary condition within the posterior distribution assembled in section 3.1. The uncertainty of heat flow increases from $\pm\,10\,\mathrm{mW\,m^{-2}}$ for shallow Curie depths $\sim 15$ km, to $\pm\,80\,\mathrm{mW\,m^{-2}}$ for Curie depths $> 40$ km (Figure 9b). Our method for determining surface heat flow from Curie depth is, therefore, most reliable in hot regions. Currently, uncertainty does not incorporate any variation in thermal properties assigned to each crustal layer or the geometry of each layer above the lower boundary condition. Uncertainty would likely increase if different thermal properties and layer geometries were considered in the ensemble of heat flow simulations, but they are not included here because already the uncertainty for Curie depths $> 30$ km is more than the range of physically plausible heat flow. In these locations, assimilating alternate geophysical sources, such as seismic data (i.e. refraction, surface waves, or receiver functions), may offer better constraints on geothermal heat flow.

Thinning of the lithosphere towards the north of the ISZ has been associated with a branch of the Icelandic mantle plume (e.g. Landes et al., 2007; Al-Kindi et al., 2003; Kirstein and Timmerman, 2000). The lithosphere-asthenosphere boundary (LAB) has been shown to be 30 km shallower in the northern part of Ireland (Landes et al., 2007; Fullea et al., 2014; Baykiev et al., 2018), which is interpreted to be a result of the spreading head of the early Icelandic plume (Landes et al., 2007). A similar degree of thermal erosion, interpreted from seismic tomography data, appears to continue to the west coast of Scotland (Arrowsmith et al., 2005). This, in conjunction with Cenozoic volcanism and thin crust (20–27 km) in northern Britain and Ireland, suggests that this region is underlain by magmatic under-plating (Davis et al., 2012). The shallow Curie depths we estimate ($\sim 15$ km) and the associated simulated heat flow, in excess of $90\,\mathrm{mW\,m^{-2}}$, lends further evidence that higher crustal temperatures are present in northern Britain and Ireland, possibly suggesting processes related to the Icelandic plume.



# 5   Conclusions

The Curie depth, often interpreted as the depth to $580\,^\circ\mathrm{C}$, offers a valuable constraint to estimate geothermal heat flow at a large scale. Spectral methods estimate Curie depth from the slope of the radial power spectrum across a square window of the magnetic anomaly. While these methods are commonly used, they are subjective to the interpreter and underestimate the uncertainty of Curie depth estimates. We cast the analytical expression of the power spectrum formulated in Bouligand et al. (2009) within a Bayesian framework, where parameters relating to the thickness of magnetic sources are expressed in probabilistic terms.

The uncertainty of Curie depth increases rapidly with depth but can be tempered with larger window sizes. Windows of the magnetic anomaly need to be 15–30 times larger than the deepest possible magnetic base in the study area to resolve precise Curie depth estimates. In the British Isles, the Curie depth is broadly delineated by the SW–NE Iapetus Suture, which separates Laurentian and Avalonian continental blocks. The uncertainty exponentially increases with Curie depth, which is due to fewer points in the low wavenumber range of the radial power spectrum. Curie depths of $> 40$ km correspond to uncertainties $\pm\,20$ km using a window size of 800 km. At these depths, the skew of the posterior distribution is used to indicate which window sizes are required to resolve Curie depth.

Surface heat flow is estimated by simulating the temperature distribution across a stratified model of the crust from the surface to the Curie depth. This 3D model incorporates sedimentary thickness and vertically partitioned heat-producing elements. The simulated heat flow matches heat flow data to a reasonable degree of uncertainty, except where igneous bodies outcrop. Granitic intrusions can contribute significantly to surface heat flow due their high rates of heat production, and heat refracts around mafic volcanics because of their low thermal conductivity. These results lend further support to lithospheric thinning in North Britain and Ireland due to the presence of the proto-Icelandic mantle plume.

*Code and data availability.*   All data presented in this article may be obtained in the supporting information from the online version, and may be reproduced from algorithms implemented within PyCurious: an open source Python package to compute Curie depth from the magnetic anomaly, https://www.github.com/brmather/pycurious.

## Appendix A:  MCMC sampling

The commonly used Metropolis-Hastings algorithm samples the posterior distribution, $P(\mathbf{m}|\mathbf{d})$, using a random walk where $k = 1, 2, \dots, n$ for a Markov Chain $n$ samples long. The random walk is initialised at a random point in the prior distribution and progresses in accordance to the following algorithm:

1. Generate a proposal $\mathbf{m}'$ for the next sample by picking from the prior distribution $P(\mathbf{m})$.

2. Calculate the acceptance ratio between each sample of the posterior $\alpha = P(\mathbf{m}'|\mathbf{d})/P(\mathbf{m}_k|\mathbf{d})$

3. Generate a random number, $0 \leq u \leq 1$, which will be used to decide if $\mathbf{m}'$ is accepted or rejected.





1        (a) accept if $u \leq \alpha$ and set $\mathbf{m}_{k+1} = \mathbf{m}'$

2        (b) reject if $u > \alpha$ and set $\mathbf{m}_{k+1} = \mathbf{m}_k$ instead

Numerous chains were initiated to ensure they converged to a similar solution.
*Acknowledgements.* This work was made possible by the G.O.THERM.3D project, supported by an Irish Research Council Research for
Policy & Society grant (RfPS/2016/50) co-funded by Geological Survey Ireland and by Sustainable Energy Authority Of Ireland. J. Fullea
was supported by a by Science Foundation Ireland grant iTHERC (16/ERCD/4303)



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
