# Peer review of "Constraining the geotherm beneath the British Isles from Bayesian inversion of Curie depth: integrated modelling of magnetic, geothermal and seismic data"

_Solid Earth, 2019_

## Referee Comment (RC1) · Anonymous Referee #1 · 20 Mar 2019

This paper uses estimates of the depth to the Curie isotherm from a spectral analysis of magnetic anomalies to model a temperature distribution of the crust from which a simulated heat flow field is created. Its novelty is in applying different window sizes for the estimation of Curie depths and using statistical techniques to quantify uncertainty. The methodologies are explained succinctly and applied methodically.

In general the Curie depth estimates are commensurate with crustal thicknesses estimated from geophysical techniques. Deeper Curie depths have higher uncertainty and therefore it is unclear how reliable the results of the Curie depths in southwest Ireland

are. There is a discussion on the effect of increasing window sizes, but does this mean these results should be discounted due to the large error? Similarly very deep Curie depths are shown in the southern North Sea. I have assumed this is an error due to a lack of magnetic anomalies.

The heat flow map is based on a crustal model by Baykiev et al. (2018) and predicts increased heat flow in northwest Scotland, in excess of 90 mW m-2. Heat flow of this magnitude is normally associated with crustal regions enriched in radiogenic minerals. There is a lack of surface heat flow measurements to corroborate the simulated heat flow, but some are shown for the tip of northeast Scotland. These measured data are less than the simulated heat flow data. Does this imply that the simulated heat flow is too high? If not, why do the data disagree? The measured heat flow are from Pollack et al. (1993) which is an abstract. As the data cannot be referenced they should be included in the paper and compared to the simulated data.

---

## Referee Comment (RC2) · Jian Wang (Referee) · 27 Mar 2019

This manuscript presents a new technique to quantify the uncertainty of Curie depths (and heat flow) based on the Bayesian framework. The authors describe the method clearly and perform the numerical synthetic tests on the effects of window sizes, fractal exponent, top and extent depths of magnetic sources. However, my major concern for this manuscript is the evaluation of the uncertainties of both the Curie depths and the predicted heat flow. Before publication it requires some modifications and clarifications. I recommend to better describe the synthetic tests and their parameters and extend the

discussion of the uncertainties. All unclear points are indicated below.

General comments:

1. In the synthetic tests part (section 2.3), the uncertainty ups to ~11 km using the window size of 200*200 km2 (Fig. 2c) which is usually used to estimate Curie depths in global and regional scales. This indicates that the relative uncertainty is ca. 100% and even 200% when the window size reduces to 100*100 km2. Such uncertainties, for me, are too large to be acceptable. For the real magnetic data, the relative uncertainty of Curie depth is ~ 50% in the southern North Sea (Fig. 6). Accordingly, the relative uncertainty of the predicted heat flow ups to ~100% (?) in the same region (Fig. 9). How to evaluate these large uncertainties. Recently, Arnaiz-Rodríguez and Orihuela (2013), Speranza et al. (2016) and Wang and Li (2018) estimated that most of the relative errors are less than 15% using various magnetic data. Their error estimation is based on the standard deviation between the spectrum and the linear fit (Okubo and Matsunaga, 1994). Wang and Li (2015, 2018), Li and Wang (2016, 2018), Li et al. (2017) applied the average values of Curie depths estimated by different window sizes to further reduce the uncertainties.

2. What's the value of beta used for estimating the Curie depth in this study? Does it vary through the study area as the window sizes shown in Fig. 5? If this is the case, the authors should provide a map of various beta for comparing with Curie depths. I noted that a larger Dz can be compensated by a smaller beta (Fig. 3c). If the authors employ variable beta, the large Curie depth/uncertainty may be caused by improper beta.

Specific comments listed as Page No.-Line No.:

1. P4-L15: In the centroid method (Tanaka et al., 1999), the wavenumber range is critical for the centroid depth (and therefore the Curie depth) estimation. Although the wavenumber segment selections vary in different studies, most researches took the wavenumber ranges less than ~0.05 km-1 (See Appendix in Wang and Li, 2018).

2. Eq (2): Please double check the fourth term, beta or beta-1.

3. P4-L22: It is difficult to estimate all the three unknown parameters simultaneously by nonlinear fitting the radial power spectrum. Bouligand et al. (2009) used a constant beta of 3.0 to obtain the Curie depth. I don't know what the value of beta is used in this study as mentioned in the above General comments. Li et al. (2013) demonstrated that the Maus and Blakely models of radial amplitude spectrum are nearly identical in shapes except for a vertical constant shift, and both are applicable to Curie depth estimation in using the centroid method.

4. Fig. 2b: Please provide the inverted parameters for different window sizes on the figure or in the figure caption.

5. Fig. 5: Please plot the centers of each window on the map.

6. Fig. 7a: Please provide the beta value on the figure and in the caption.

Recommended new references:

1. Arnaiz-Rodríguez, M.S., Orihuela, N., 2013. Curie point depth in Venezuela and the Eastern Caribbean. Tectonophysics 590, 38-51.

2. Speranza, F., Minelli, L., Pignatelli, A., Gilardi, M., 2016. Curie temperature depths in the Alps and the Po Plain (northern Italy): Comparison with heat flow and seismic tomography data. J Geodyn 98, 19-30.

3. Okubo, Y., Matsunaga, T., 1994. Curie point depth in northeast Japan and its correlation with regional thermal structure and seismicity. Journal of Geophysical Research: Solid Earth 99, 22363-22371.

4. Jian Wang, Chun-Feng Li. 2015. Crustal magmatism and lithospheric geothermal state of western North America and their implications for a magnetic mantle. Tectonophysics, 638: 112-125.

5. Jian Wang, Chun-Feng Li. 2018. Curie point depths in Northeast China and their

geothermal implications for the Songliao Basin. Journal of Asian Earth Sciences, 163: 177-193.

6. Chun-Feng Li, Jian Wang. 2018. Thermal structures of the Pacific lithosphere from magnetic anomaly inversion. Earth and Planetary Physics, 2: 52-66.

7. Chun-Feng Li, Jian Wang. 2016. Variations in Moho and Curie depths and heat flow in Eastern and Southeastern Asia. Marine Geophysical Research, 37 (1): 1-20.

8. Chun-Feng Li, Jian Wang, Jian Lin, Tingting Wang. 2013. Thermal evolution of the North Atlantic lithosphere: New constraints from magnetic anomaly inversion with a fractal magnetization model. Geochemistry Geophysics Geosystems, 14 (12): 5078-5105.

Best regards

Dr. Jian Wang

―――――――――――――――――――――

---

## Author Comment (AC1) · 8 May 2019

We would like to thank the reviewer for the constructive comments and suggestions. Please find a point-by-point response to each of the issues you have raised in your review.

(1) Deeper Curie depths have higher uncertainty and therefore it is unclear how reliable the results of the Curie depths in southwest Ireland are. There is a discussion on the effect of increasing window sizes, but does this mean these results should be

discounted due to the large error?

The reviewer has highlighted an important observation that the uncertainty of Curie depth increases substantially with depth. Based on our results, we find that the method of Curie depth calculation (explained in the methods section) is generally only suitable to quantitatively resolve relatively shallow magnetic layers. The presence of deep Curie Depths should be regarded as a quantitative feature from the inversion, i.e. a likely cold lithospheric region where the amplitude (temperature in our case) is not well resolved. Whether the highly uncertain results in SW Ireland should be discounted is up to interpretation and is something that, in any case, would require considering additional independent data sets. The latter would be out of the scope of the present paper where we rather focus on the degree of uncertainty associated with Curie depth estimates, in line with the topic of the special issue "Understanding the unknowns: the impact of uncertainty in the geosciences".

(2) Similarly very deep Curie depths are shown in the southern North Sea. I have assumed this is an error due to a lack of magnetic anomalies.

This is a good point. Our Curie depth estimations use the EMAG2 magnetic anomaly, which splices multiple regional grids and satellite data. The effective resolution of these data vary spatially. On the matter of spatial resolution of EMAG2 we add "It is important to note that the effective resolution of this global compilation is inherited from multiple regional grids and satellite data that are spliced together to form the EMAG2 dataset." On the matter of uncertainty we add "Higher uncertainty in the southern North Sea is compounded by the lack of magnetic data and thus very large window sizes in order to capture any sensitivity to the magnetic thickness."

(3) There is a lack of surface heat flow measurements to corroborate the simulated heat flow, but some are shown for the tip of northeast Scotland. These measured data are less than the simulated heat flow data. Does this imply that the simulated heat flow is too high? If not, why do the data disagree?

The misfit between the simulated heat flow data and measured data are 5-10mW/m2 in northeast Scotland. These differences could be attributed to local effects perturbing the regional geotherm. We find that this is quite low compared to the large misfits observed within granitic batholiths (indicated by white circles overlain on the heat flow map – Figure 9a) because of substantial upper crustal enrichment in heat-producing elements. To clarify our position on this matter we have added the following text to the discussion section of the manuscript, "In general, Curie depth estimates are sensitive to the regional heat flow regime and cannot resolve anomalies that locally alter surface heat flow. These effects include granitic intrusions and hydrothermal advection among others. In spite of this, the misfit between simulated heat flow and data do not exceed one standard deviation of all thermal models in the ensemble (Figure 9b). In locations of high misfit, assimilating alternate geophysical sources, such as seismic data (i.e. refraction, surface waves, or receiver functions), may offer better constraints on geothermal heat flow." On the matter of the spatial distribution of heat flow data, we add "Heat flow data is clustered mainly within coastlines and in some localised areas offshore." We outline the misfit between data and observations as follows:

(4) The measured heat flow are from Pollack et al. (1993) which is an abstract. As the data cannot be referenced they should be included in the paper and compared to the simulated data.

Thank you to the reviewer for bringing this to our attention. The custodians of these data seems to be replaced reguarly. The most recent location is this website: https://engineering.und.edu/research/global-heat-flow-database/data.html which we embed within within the original reference for Pollack et al 1993.

---

## Author Comment (AC2) · 8 May 2019

We would like to thank the reviewer for the constructive comments and suggestions. Please find our response to each of the issues you have raised below.

General comments

A general observation by the reviewer is that Curie depths, in the synthetic tests and real magnetic data, are highly uncertain. The reason for this is because we allow beta to vary across the study area. It is common practise in most of the literature to fix the

fractal parameter, beta, to a constant value. Implicit in this decision is to assume that the magnetic composition of the rocks is consistant at long wavelengths – an assumption that is less justified on continents than oceanic crust. As Bouligand et al. (2009) note: "the fractal parameter beta, which is the slope of the power spectrum in a log-log scale, is related to the geology and thus might vary geographically depending on rock types or geologic structures." We showed in our synthetic tests that the thickness of magnetic sources, dz, and beta are strongly correlated, as noted by the reviewer, thus a fixed value of beta reduces the amount of variation of dz and underestimates the uncertainty of each Curie depth determination. This explains the much higher degree of uncertainty in our Curie depth determinations compared to many other studies. Furthermore, the motivation to fix beta is problematic within a Bayesian framework where each of the parameters that control Curie depth are expressed probabilistically. In our opinion the latter arguments reasonably justify our decision to allow beta to vary across the study area. We have made this more explicit in the text where we add "Most studies fix beta to a constant value across the entire study area, but implicit in this decision is to assume that the magnetic composition of the rocks is consistent over long wavelengths. Through casting beta as an inversion variable, we retrieve a comparitively higher degree of uncertainty because our method propagates all of the errors associated with each parameter within a Bayesian framework. If this parameter is not taken into account then it is likely that all other parameters (zt, dz, and C) will be biased."

—

What's the value of beta used for estimating the Curie depth in this study? Does it vary through the study area as the window sizes shown in Fig. 5? If this is the case, the authors should provide a map of various beta for comparing with Curie depths. I noted that a larger Dz can be compensated by a smaller beta (Fig. 3c). If the authors employ variable beta, the large Curie depth/uncertainty may be caused by improper beta.

Thank you for the suggestion to include a map of the variation in beta across the study area. We have included it as a subfigure (5b) adjacent to the map of window sizes

(5a). Please refer to the general discussion above for our motivations in allowing beta to vary across the study area.

Specific comments

1. P4-L15: In the centroid method (Tanaka et al., 1999), the wavenumber range is critical for the centroid depth (and therefore the Curie depth) estimation. Although the wavenumber segment selections vary in different studies, most researches took the wavenumber ranges less than âĹij0.05 km-1 (See Appendix in Wang and Li, 2018).

This is a good point, to clear up the confusion in this section we have added the following text: In this method of Curie depth estimation, the wavenumber segments over which to calculate the depth of magnetic sources varies in different studies, but is usually <0.05 km-1 (Wang and Li, 2018).

2. Eq (2): Please double check the fourth term, beta or beta-1.

Fixed. Thanks for noticing this difficult-to-spot typo!

3. P4-L22: It is difficult to estimate all the three unknown parameters simultaneously by nonlinear fitting the radial power spectrum. Bouligand et al. (2009) used a constant beta of 3.0 to obtain the Curie depth. I don't know what the value of beta is used in this study as mentioned in the above General comments. Li et al. (2013) demonstrated that the Maus and Blakely models of radial amplitude spectrum are nearly identical in shapes except for a vertical constant shift, and both are applicable to Curie depth estimation in using the centroid method.

Please refer to the general discussion above for our motivations in allowing beta to vary across the study area. Bouligand et al. (2009) fixed beta to a constant of 3.0 because it resulted in the lowest misfit out of all their forward models, but they found regions of their study area where the misfit between the radial power spectrum and their fitted curve were still very high, and would be better fitted with a different beta. There is no such problem here, because beta is allowed to vary for every centroid

so the nonlinear fit to the radial power spectrum is always optimal. We have added the following comparison between both methodologies, "Li et al (2013) demonstrated that the radial power spectrum of Maus et al. (1997) and Bouligand et al. (2009) are nearly identical, except for a constant vertical shift. Because the thickness of a buried magnetic layer depends only on the slope of the power spectrum, they yield equivalent results."

4. Fig. 2b: Please provide the inverted parameters for different window sizes on the figure or in the figure caption.

Done! We have included the inverted parameters within the figure caption.

5. Fig. 5: Please plot the centers of each window on the map.

Done!

6. Fig. 7a: Please provide the beta value on the figure and in the caption.

We have included a map of the variation in beta (Figure 5b) and included this parameter in the supplementary data table.

---

## Editor Decision (ED1)

Original version

New version

(a) Window size

300    400    500    600    700
Window size

200   300    400    500    600    700
Window size (km)

---

## Author Response (AR2)

**Response to editor comments (2[nd] revision)**

Thank you for your constructive comments of our manuscript. We answer the comments in blue below, and provide a revised version of the paper with changes indicated in the text.

1) Figure 5a is sensibly different than Figure 5 in the first submission, not only because of the addition of the centroids of the windows (see the attached document). In the original figure, there seemed to be some sort of interpolation between the different windows, whereas here there is no overlap between the windows (I'm assuming they are 25x25km; please specify in the caption). Was this just a mistake or there was something different going on within the processing flow used?

> Thank you for pointing this out. In the last iterations of the manuscript we clearly omitted the details of the spatial resolution we have used in processing Curie depth. The change in Figure 5a is observed because we turned off interpolation of the grid, which we believe provides a more suitable context from which to interpret all subsequent figures that are interpolated. We have added the following sentences in the caption of Figure 5 to clear up this confusion: "The ensemble of Curie depth estimates were computed across centroids spaced 20x20km apart, as indicated by black dots in each sub-figure. (a) Combination of window sizes used to generate the map of Curie depth (Figure 6), each pixel illustrates the spatial resolution of our results; (b) maximum *a posteriori* distribution of beta inverted from the magnetic anomaly and interpolated to the map resolution of 0.01°." In addition to "The centroid spacing was set to 20x20 km eastings and northings, respectively" added to the start of the results section.

2) At the end of section 3.2, you associate part of the differences between your heat flow model and the heat flow observations to the insulating properties of the sedimentary cover in some areas. Could you add a (small) map in plain view (potentially in Figure 8) featuring the sediment bodies and the Laurentia/Avalonia boundary? I think this would be helpful to compare with Figure 9 and to support the discussion.

> We have added an additional figure (8), which adds the sedimentary thickness and crustal thickness from Baykiev et al. (2018): "Maps of layer geometries extracted from the model of Baykiev *et al.* (2018), overlain with major tectonic structures. (a) Sedimentary thickness and (b) crustal thickness variations across the British Isles. The Iapetus Suture marks the boundary between Laurentia and Avalonia." We agree this makes interpretation of the heat flow map (now figure 10) clearer for the reader.

Thank you again for your positive reviews of our manuscript.

Yours sincerely,

Ben Mather (on behalf of all authors)

[revised manuscript text omitted]